# Viral Inactivation by Light-Emitting Diodes: Action Spectra Reveal Genomic Damage as the Primary Mechanism

**DOI:** 10.3390/v17081065

**Published:** 2025-07-30

**Authors:** Kazuaki Mawatari, Yasuko Kadomura-Ishikawa, Takahiro Emoto, Yushi Onoda, Kai Ishida, Sae Toda, Takashi Uebanso, Toshihiko Aizawa, Shigeharu Yamauchi, Yasuo Fujikawa, Tomotake Tanaka, Xing Li, Eduardo Suarez-Lopez, Richard J. Kuhn, Ernest R. Blatchley, Akira Takahashi

**Affiliations:** 1Department of Preventive Environment and Nutrition, Institute of Biomedical Sciences, Tokushima University Graduate School, Tokushima 770-8503, Tokushima, Japan; ishikawa-kd@tokushima-u.ac.jp (Y.K.-I.); yushi.onoda@nichia.co.jp (Y.O.); kisd29@koto.kpu-m.ac.jp (K.I.); c202441002@tokushima-u.ac.jp (S.T.); uebanso@tokushima-u.ac.jp (T.U.); akiratak@tokushima-u.ac.jp (A.T.); 2Department of Microbial Control, Institute of Biomedical Sciences, Tokushima University Graduate School, Tokushima 770-8503, Tokushima, Japan; 3Graduate School of Science and Technology, Tokushima University, Tokushima 770-8506, Tokushima, Japan; emoto@tokushima-u.ac.jp; 4Nichia Corporation, Anan 774-8601, Tokushima, Japan; toshihiko.aizawa@nichia.co.jp (T.A.); shigeharu.yamauchi@nichia.com (S.Y.); yasuo.fujikawa@nichia.co.jp (Y.F.); tomotake.tanaka@nichia.co.jp (T.T.); 5Lyles School of Civil & Construction Engineering, Purdue University, West Lafayette, IN 47907, USA; li1856@purdue.edu (X.L.);; 6Biological Sciences, Purdue University, West Lafayette, IN 47907, USA; esuarezl@purdue.edu (E.S.-L.); kuhnr@purdue.edu (R.J.K.); 7School of Sustainability Engineering & Environmental Engineering, Purdue University, West Lafayette, IN 47907, USA

**Keywords:** UV-LED, action spectrum, enveloped RNA virus, nonenveloped virus, protein, nucleic acid

## Abstract

Irradiation with ultraviolet light-emitting diodes (UV-LEDs) represents a promising method for viral inactivation, but a detailed understanding of the wavelength-dependent action spectra remains limited, particularly across different viral components. In this study, we established standardized UV action spectra for infectivity reduction in pathogenic viruses using a system equipped with interchangeable LEDs at 13 different peak wavelengths (250–365 nm). The reduction in viral infectivity induced by UV-LED exposure was strongly related to viral genome damage, whereas no significant degradation of viral structural proteins was detected. Peak virucidal efficiency was observed at 267–270 nm across all tested viruses, representing a slight shift from the traditionally expected 260 nm nucleic acid absorption peak. Enveloped RNA viruses, including influenza A virus, respiratory syncytial virus, and coronavirus, exhibited greater UV sensitivity than nonenveloped viruses such as feline calicivirus and adenovirus. These observations indicate that structural characteristics, such as the presence of an envelope and genome organization, influence UV susceptibility. The wavelength-specific action spectra established in this study provide critical data for optimizing UV-LED disinfection systems to achieve efficient viral inactivation while minimizing energy consumption in healthcare, food safety, and environmental sanitation.

## 1. Introduction

Viruses are microscopic infectious agents that require a host cell to replicate [1]. Viruses carry genetic material (either DNA or RNA) enclosed within the capsid [2]. Some viruses also possess an envelope, a lipid bilayer derived from the host cell membrane, which contains viral surface proteins (e.g., spike and glycoproteins) that facilitate host cell entry and immune evasion [3]. Viruses propagate within specific host cells by utilizing the host’s complex metabolic and biosynthetic machinery. The viral genome is packaged within the capsid, which together with nucleic acid-associated proteins forms the nucleocapsid [2]. After attachment, viruses deliver their genetic material into the host cell, hijack its internal machinery to replicate, and eventually disrupt the cell to release newly formed viral particles [4]. Virus propagation is a multistep process that enables viruses to spread and infect other cells. This process begins with attachment to specific cell surface receptors, followed by entry via membrane fusion or endocytosis [4,5]. Once inside, viral DNA or RNA is released, and it is replicated and transcribed using host or viral enzymes. Newly synthesized viral genomes and proteins are assembled into virions that are subsequently released through cell lysis or budding [6]. These virions can infect additional cells, continuing the propagation cycle.

Virus transmission occurs through various pathways, facilitating the spread of infections among hosts [7,8]. Common transmission routes include direct contact (e.g., person-to-person and bodily fluids), airborne (e.g., respiratory droplets and aerosols), vector-borne (e.g., mosquitoes and ticks), and fomite transmission (e.g., contaminated surfaces, water, and food) [8,9]. Enveloped viruses are generally more susceptible to environmental factors such as heat, desiccation, and disinfectants, whereas nonenveloped viruses tend to exhibit greater environmental stability [10,11].

Virus disinfection methods are critical for preventing the spread of infections in healthcare, industrial, and public environments [9,12]. Effective disinfection strategies target key viral components, including the envelope, capsid, and nucleic acids. Common approaches include chemical disinfectants (e.g., alcohol, bleach, and hydrogen peroxide), biological agents (e.g., antiviral peptides, and enzymes), and non-chemical methods (e.g., heat, ultraviolet [UV] irradiation, photocatalysis, high pressure, and desiccation). The optimal disinfection method depends on the virus type, environmental conditions, and surface materials [11].

Physical methods leverage environmental factors to disrupt viral structures and eliminate infectivity. Heat and dry-heat treatments denature viral proteins, but they are unsuitable for raw foods or non-thermotolerant materials; therefore, processes must be optimized to preserve food quality [13,14]. UV radiation damages viral nucleic acids, preventing replication [15,16]. Photocatalytic effects induced by UV- or visible-light irradiation of photocatalytic materials (e.g., TiO_2_) primarily degrade viral proteins [17]. High-pressure treatment compromises viral integrity, whereas desiccation reduces stability by removing essential moisture [18]. These chemical-free methods offer broad-spectrum effectiveness, and they are widely applicable in healthcare, food safety, and sterilization settings [11]. Understanding these mechanisms is crucial for optimizing virus inactivation strategies for public health protection.

UV disinfection is a highly effective method for virus inactivation that primarily targets viral DNA or RNA to inhibit replication [19,20]. Low-pressure UV systems, which are characterized by a primary emission line at 253.7 nm, have been extensively used to disinfect water, air, and surfaces [15,21,22]. UV radiation is classified by wavelength into UVA (315–400 nm), UVB (280–315 nm), and UVC (<280 nm). Among these, UVC is the most effective for viral inactivation because it directly damages viral nucleic acids [23]. Conversely, there is limited evidence supporting the significant virucidal effects of UVB and UVA, as these wavelengths less efficiently disrupt viral genomes [20,24,25]. Although UVB and UVA can contribute to virus inactivation, they require significantly higher doses than UVC. In particular, UVA primarily induces indirect oxidative damage through the generation of reactive oxygen species [26,27,28].

In recent years, UV light-emitting diodes (LEDs) have emerged as alternative UV sources. UV-LEDs can emit light at specific wavelengths without requiring optical filters, and they can be designed to emit across a wide range of wavelengths, thereby improving flexibility and energy efficiency in disinfection applications [29]. Recent advances in semiconductor technology have enabled precise wavelength control in UV-LEDs, enhancing disinfection efficiency across healthcare, water treatment, and air sterilization applications [29].

We previously demonstrated that irradiation with 310 nm UVB-LEDs at a dose of 0.26 J/cm^2^ and 365 nm UVA-LEDs at a dose of 6.4 J/cm^2^ reduced the infectivity of influenza A virus (IAV) by 90%, likely by inhibiting the replication and transcription of viral RNA within host cells [20]. The development of AlGaN-based semiconductor materials has significantly improved the performance of UVC-LEDs, enabling higher efficiency and stability [30]. UVC radiation inactivates viruses primarily by directly damaging nucleic acids through the formation of pyrimidine dimers, which disrupt replication and lead to viral inactivation [25,31]. Tanaka et al. reported that irradiation with 255 nm UV-LEDs at a fluence of 20 mJ/cm^2^ reduced the infectivity of nonenveloped feline calicivirus by 3 log_10_, decreased viral RNA levels by 43%, and reduced capsid protein expression by 6.9% [32]. Irradiation with 281 nm UV-LEDs achieved similar reductions in infectivity and viral RNA but induced a greater decrease in capsid protein expression (16.2%) than 255 nm UV-LEDs. Additionally, Beck et al. demonstrated that high doses (400 mJ/cm^2^) delivered by 261 and 278 nm UV-LEDs reduced protein expression by 11–34% and 17–20%, respectively, in nonenveloped human adenovirus 2 (AdV-2) [33]. Minamikawa et al. reported that irradiation with 300 nm UV-LEDs at a fluence of 23 mJ/cm^2^ reduced the infectivity of severe acute respiratory syndrome coronavirus 2 (SARS-CoV-2) by 4 log_10_ [34]. Similarly, Oguma et al. found that irradiation with 300 nm UV-LEDs at 600 mJ/cm^2^ reduced the infectivity of feline calicivirus by 4 log_10_ [35].

Although research has established the efficacy of UVC-LEDs in reducing viral infectivity by damaging viral RNA and structural proteins, precise action spectra are required to optimize their application in healthcare, water treatment, and public safety. Viral sensitivity to UV-LED irradiation varies across strains, but direct comparisons remain challenging because of differences in LED wavelengths and experimental conditions. To overcome these limitations and support broader UV-LED implementation, it is essential to establish accurate viral action spectra that describe the wavelength-dependent sensitivity of viruses to UV irradiation.

To accurately compare and evaluate UV sensitivity across different viral species, it is crucial to standardize irradiation-related conditions, such as peak wavelength, irradiance, beam angle, bottom surface reflectivity, stirrer usage, sample temperature, and sample volume, during exposure. We recently developed an irradiation system and established a standardized evaluation method for assessing the antimicrobial effects of LED irradiation [36,37,38]. The system permits interchangeable LEDs with peak emission at 13 different wavelengths (250 [U250], 254 [U254], 257 [U257], 260 [U260], 263 [U263], 267 [U267], 270 [U270], 275 [U275], 280 [U280], 290 [U290], 300 [U300], 308 [U308], and 365 nm [U365]). Using this system, we previously elucidated the UV sensitivity profiles of bacteria and fungi [36,37,38].

In the present study, we irradiated several respiratory viruses, including coronaviruses (CoVs), IAVs, respiratory syncytial viruses (RSVs), and human metapneumovirus (HMPV), all of which are enveloped, single-stranded RNA viruses capable of aerosol transmission. Effective UV germicidal irradiation against these viruses might help control disease spread. Viral envelope proteins can be structurally or functionally altered by UV exposure at certain wavelengths, potentially contributing to viral inactivation [33,39]. We also evaluated an enveloped DNA virus (herpes simplex virus [HSV]) and nonenveloped viruses (feline calicivirus [FCV] and AdV). This study aimed to determine the action spectra of these viral strains across the UVC, UVB, and UVA regions, enabling the precise evaluation of the wavelength-specific efficacy for viral inactivation and optimizing the conditions for different virus types and environmental applications.

## 2. Materials and Methods

### 2.1. Virus and Host Cell Strains

The virus and host cell strains, along with their respective providers, are listed in Table 1. Host cells were cultured in Dulbecco’s Modified Eagle’s Medium (DMEM) supplemented with 10% fetal bovine serum (Thermo Fisher Scientific, Waltham, MA, USA) and 50 µg/mL gentamicin (Fujifilm Wako Chemicals, Osaka, Japan) at 37 °C in an atmosphere of 5% CO_2_. IAVs were propagated in 10-day-old embryonated chicken eggs (Ishii Poultry Agricultural Co., Tokushima, Japan) for 48 h at 37 °C [20,24]. The other viruses were propagated in their respective host cells for 48–84 h at 37 °C. The chorioallantoic fluids from the eggs and the culture supernatants from the cells were precleared via centrifugation at 3300× *g* for 5 min, followed by filtration through 0.22-µm membrane filters. The clarified supernatants were then collected and stored at −80 °C until use in the LED irradiation experiments.

### 2.2. Irradiation of Viral Suspensions by LEDs

We previously developed a standardized UV-LED irradiation system and reported the action spectra of bactericidal and fungicidal effects using this platform [36,37,38]. To enhance collimation and minimize stray radiation, the system body incorporates three apertures coated with a low-reflectance material (Appendix A). The layout of the 36 LED chips on the circuit board was optimized through optical simulations to ensure high uniformity of the collimated radiation directed at the viral suspension in a Petri dish. To improve the stability and reproducibility of both the fluence rate and peak wavelength, the LED-mounted circuit board was placed on a heat sink to maintain a consistent temperature. The fluence rate of each LED was measured using a radiometer calibrated by the Japan Calibration Service System, and the measured values were confirmed to be within a 10% error margin. Viral suspensions were irradiated using LEDs with 13 different peak wavelengths (U250, U254, U257, U260, U263, U267, U270, U275, U280, U290, U300, U308, and U365; Appendix A). The actual peak wavelengths and corresponding fluence rates at the sample surface for each LED are provided in Appendix A. Each viral suspension was diluted in phosphate-buffered saline, and 1 mL of the prepared suspension was dispensed into a 35 mm Petri dish for irradiation. Viral suspensions were irradiated by UV-LEDs up to 60 mJ/cm^2^ in the U280-LED experiments, or up to 10 mJ/cm^2^ in the experiments using the other 13 LEDs. Except for the U250-, U290-, and U365-LEDs, the fluence rates of most UV-LEDs were adjusted to exactly 1 mW/cm^2^, meaning that a fluence of 10 mJ/cm^2^ was achieved with 10 s of irradiation. The fluence rates of the U250-, U290-, and U365-LEDs were 0.5, 1.5, and 18.0 mW/cm^2^, respectively; therefore, a fluence of 10 mJ/cm^2^ was achieved in 20, 6.7, and 0.6 s, respectively.

### 2.3. Measurements of Virus Infectivity

A plaque-forming unit (PFU) assay was conducted using host cells to evaluate the cytopathic effects of each virus. After LED irradiation of the viral suspensions, cells cultured in DMEM in six-well plates were infected in duplicate with 10-fold serial dilutions of the irradiated suspensions and incubated at 37 °C for 48–84 h. An unirradiated viral suspension served as the dark control. After incubation, the cells were fixed and stained with a 0.5% crystal violet solution containing 10% formalin for PFU quantification (Appendix A). The viral inactivation efficiency of UV-LED irradiation was assessed by calculating the log_10_ reduction in infectivity as follows: log PFU/mL ratio = log_10_(*Nt*/*N*_0_), where *Nt* is the PFU/mL of the UV-irradiated sample and *N*_0_ is the PFU/mL of the non-irradiated control. The inactivation rate constant *k* was calculated as −ln[*Nt*/*N*_0_] divided by the fluence (mJ/cm^2^) for each wavelength based on the assumption of first-order (single-event) inactivation kinetics.

### 2.4. RNA Extraction and Reverse Transcription Quantitative Real-Time Polymerase Chain Reaction (RT-qPCR)

Damage to viral RNA for the four RNA viruses, IAV H1N1, SARS-CoV-2, RSV strain long, and HMPV, caused by UV irradiation was assessed using RT-qPCR. Viral RNA was extracted from UV-irradiated or control suspensions using the High Pure Viral RNA Kit (Roche, Mannheim, Germany) following the manufacturer’s instructions. Strand-specific reverse transcription of viral RNA was performed using the PrimeScript RT reagent (TaKaRa, Kusatsu, Japan) using the primers listed in Appendix A based on a modified protocol previously described [20]. Real-time PCR was conducted using TB Green Fast qPCR (TaKaRa) on a CFX Duet real-time PCR system (Bio-Rad Laboratories, Hercules, CA, USA). The thermal cycling conditions were as follows: initial denaturation at 95 °C for 30 s, followed by 40 cycles of annealing at 95 °C for 5 s and extension at 60 °C for 10 s.

### 2.5. Dot-Blot Analysis

Damage to HSV-1 viral DNA caused by UV irradiation was assessed using dot-blot analysis. Viral DNA was extracted from HSV-1 suspensions with and without UV-LED irradiation using the DNAsol Nucleic Acid Extraction Kit (Molecular Research Center, Cincinnati, OH, USA) according to the manufacturer’s instructions. The extracted viral DNA was filtered through a 0.45 µm nitrocellulose membrane (Fujifilm Wako Pure Chemicals, Tokyo, Japan) using the Bio-Dot SF Microfiltration Apparatus (Bio-Rad Laboratories). The membrane was dried at 37 °C and then blocked with 5% bovine serum albumin in Tris-buffered saline containing Tween 20 (TBS-T) for 1 h to prevent non-specific binding. Subsequently, the membrane was incubated with the anti-cyclobutane pyrimidine dimer (CPD) antibody (Cosmo Bio, Tokyo, Japan) at 4 °C for 12 h. After three washes with TBS-T, the membrane was incubated with the horseradish peroxidase-conjugated anti-mouse immunoglobulin antibody (MBL, Nagoya, Japan) for 1 h. Immunoreactive signals were visualized using an enhanced chemiluminescence detection system (Amersham Biosciences, Buckinghamshire, UK). Images were captured using the LAS-3000UV mini CCD camera system (Fujifilm) and analyzed densitometrically with the ImageJ 1.53a software (US National Institutes of Health, Bethesda, MD, USA).

### 2.6. Western Blotting

Following either UV-LED irradiation or no irradiation, the viral suspensions were homogenized with the lysis buffer, and protein expression was assessed by Western blotting as previously described [25]. The anti-SARS-CoV-2 spike and anti-RSV nucleoprotein antibodies were obtained from Sino Biological (Nanjing, China). The anti-IAV non-structural protein (NS1) and anti-HSV-1 infected cell protein 0 (ICP0) antibodies were purchased from Santa Cruz Biotechnology (Dallas, TX, USA).

### 2.7. Statistical Analysis

Statistical analysis was performed by ANOVA followed by Bonferroni’s multiple comparison test using the StatView 5.0 software (SAS Institute Inc., Cary, NC, USA). Student’s *t*-test was applied to paired data where appropriate. *p* < 0.05 was considered statistically significant. Pearson’s correlation coefficient (*r*) and Spearman’s rank correlation test were performed using the MATLAB R2017b software (MathWorks, Natick, MA, USA) to evaluate the association between infectivity reduction and viral genome or protein levels for each virus strain.

## 3. Results

### 3.1. Differences in U280-LED Responses Were Attributed to Viral Strains and Components

To compare UV-LED sensitivity across different virus species, we exposed IAVs, CoVs, RSVs, HMPV, HSV-1, FCV, and AdV to U280-LED irradiation, a standard setup for antimicrobial irradiation. The infectivity of all viruses was reduced in a fluence-dependent manner by approximately 3–4 log_10_ (Figure 1 and Appendix A). The fluence-dependent reduction in viral infectivity, which appeared approximately linear on a log_10_ scale (Figure 1), was consistent with first-order inactivation kinetics, supporting the use of this model to calculate the inactivation rate constants (*k*). The virucidal efficiency of U280-LED irradiation against HCoV-OC43, HCoV-229E, SARS-CoV-2, and FCV closely matched previously reported results (Table 2), supporting the utility of our system as a standardized evaluation method for assessing the virucidal effects of LED irradiation.

All enveloped, single-stranded RNA viruses, including IAVs (Figure 1A), CoVs (Figure 1B), RSVs, and HMPV (Figure 1C), exhibited higher sensitivity to U280-LED irradiation than HSV-1, FCV, and AdV-5 (Figure 1D). In particular, the nonenveloped viruses FCV and AdV-5 exhibited greater resistance to U280-LED irradiation than the enveloped viruses, with AdV-5 being the most resistant virus in this study. However, *k* (ln PFU reduction/cm^2^/mJ) revealed that the sensitivity to U280-LED irradiation varied even among the enveloped, single-stranded RNA viruses, as RSVs were the most sensitive to U280-LED irradiation among the viruses tested. These results suggest that although the presence of an envelope is a critical factor influencing UV sensitivity, additional factors among enveloped viruses also contribute to their differential sensitivity.

### 3.2. All Virus Strains Exhibited Similar Action Spectra, and U267- and U270-LEDs Displayed the Greatest Virucidal Efficiency

Next, to establish accurate viral action spectra, which describe the wavelength-dependent sensitivity of viruses to UV irradiation, we irradiated suspensions of IAVs, CoVs, RSVs, HSV-1, and FCV using a system equipped with interchangeable LEDs with 13 different peak wavelengths under the same fluence (Figure 2). All virus strains exhibited similar action spectra, with quantifiable reductions in infectivity observed following LED irradiation at U290 and lower wavelengths. Among these, U267- and U270-LEDs exhibited the highest virucidal efficiency for all viruses. However, from U263 down to U250, the virucidal efficiency decreased with decreasing wavelength, particularly in RSV and HMPV. These results suggest that the virucidal effect is wavelength-dependent, and wavelengths shorter than U263 could induce relatively less efficient viral inactivation for some enveloped RNA viruses. Interestingly, the wavelength-dependent behavior observed in Figure 2 is similar in shape to the absorbance spectrum of the genomic RNA. Although we did not measure absorbance across this range for all viruses tested, preliminary absorbance data from the extracted IAV RNA exhibited a similar trend [24]. This similarity suggests that the quantum yield for photochemical damage remains relatively constant across this wavelength range.

### 3.3. Damage to Viral Genomic RNA Was Induced by U280-LED Irradiation

We sought to identify the critical viral components contributing to the differences in UV-LED sensitivity among the enveloped viruses. First, to assess UV-LED-induced damage to the viral genome, we irradiated enveloped RNA viruses (IAVs, CoVs, RSVs, and HMPV) with U280-LEDs and extracted their viral RNA. RNA damage was evaluated by measuring the reduction in the relative abundance of viral RNA using strand-specific RT-qPCR. The relative abundance of viral RNA decreased in a fluence-dependent manner for all viruses (Figure 3A), with the reduction efficiency being significantly higher for RSVs than for the other viruses (Figure 3B). These results suggest that viral RNA damage plays an important role in the higher UV sensitivity of RSV compared with other enveloped viruses.

### 3.4. Action Spectra of Infectivity Reduction by UV-LED Irradiation Were Related to Viral RNA Damage

To investigate whether the viral action spectra of infectivity reduction were related to viral genomic RNA damage, we irradiated the suspensions of IAV, SARS-CoV-2, RSV, and HMPV using a system equipped with interchangeable LEDs at 13 different wavelengths under the same fluence (Figure 4). All virus strains exhibited similar action spectra for the reduction in relative RNA abundance, with significant reductions observed following UV-LED irradiation at wavelengths of U290 and lower, peaking around U270. These results suggest that the action spectra of infectivity reduction by UV-LED irradiation are related to viral RNA damage.

### 3.5. Action Spectra of Infectivity Reduction by UV-LED Irradiation Were Related to Viral DNA Damage

To investigate whether the viral action spectra of infectivity reduction were related to viral genomic DNA damage, as observed in RNA viruses, we irradiated suspensions of HSV-1 and extracted the viral DNA. DNA damage induced by LED irradiation was evaluated by measuring the levels of DNA photoproducts, specifically CPDs (Figure 5). CPD formation leads to conformational changes in the DNA helix and the inhibition of DNA transcription and replication [41], as observed in bacteria such as *Escherichia coli* and *Staphylococcus aureus* [28,42]. The relative abundance of viral CPDs increased in a fluence-dependent manner following U280-LED irradiation (Figure 5A). Under LED irradiation at 13 different wavelengths with a fluence of 10 mJ/cm^2^, viral CPD levels exhibited an action spectrum that was vertically symmetric with the infectivity reduction spectrum, with significant increases observed following irradiation at U290 and lower wavelengths, peaking around U270. These results suggest that the action spectra of infectivity reduction by UV-LED irradiation are related to viral DNA damage.

### 3.6. Viral Protein Degradation Was Not Induced by UV-LED Irradiation

To assess protein damage induced by UV-LED irradiation, we evaluated the expression of viral proteins from IAV, SARS-CoV-2, RSV, and HSV-1 by Western blotting (Figure 6). We detected these proteins using their respective specific antibodies. U280-LED irradiation did not alter the expression of NS1 (IAV), spike glycoprotein (SARS-CoV-2), nucleoprotein (RSV), and ICP0 (HSV-1; Figure 6) even though the LED fluences were sufficient to reduce infectivity. Next, to investigate the effects of other UV-LED wavelengths, viral suspensions were irradiated using LEDs with 13 different peak wavelengths under the same fluence conditions. Irradiation with any wavelength of LEDs did not alter protein expression in any virus (Figure 7). These results suggest that UV-LED irradiation had no significant effect on viral proteins at the fluences used in this study.

To identify the critical viral components contributing to the UV action spectra of LED irradiation for infectivity reduction, we calculated *r* between the action spectra for infectivity reduction and those for viral genome or protein damage using data from Figure 2, Figure 4 and Figure 6 for IAV, SARS-CoV-2, RSV, and HSV-1. *r* exceeded 0.93 (*p* < 0.001) for correlations between viral infectivity reduction and the levels of viral genomic damage, whereas *r* was lower for correlations between the reduction in viral infectivity and viral protein expression (Table 3). To further confirm these associations, we performed Spearman’s rank correlation tests to analyze the relationships between the reduction in viral infectivity and the viral genome or protein damage induced by each UV-LED wavelength. Viral infectivity reduction was significantly correlated (*p* < 0.001) with viral genomic damage but not with viral protein degradation (Appendix A). These results suggest that the UV action spectra for viral infectivity reduction were attributable to viral genomic damage rather than protein degradation.

## 4. Discussion

This study systematically evaluated the action spectra of viral infectivity reduction induced by UV-LED irradiation across the UVC, UVB, and UVA regions. Using a standardized multi-wavelength UV-LED irradiation system, we demonstrated that viral genomic damage, as opposed to viral protein degradation, was the primary factor responsible for UV-induced viral inactivation in RNA and DNA viruses. These findings provide important insights into the fundamental mechanisms underlying UV disinfection and highlight the need for wavelength-specific optimization of UV-LED systems.

Previous studies indicated that nucleic acids exhibit maximum absorption near U260; thus, viral inactivation was theoretically expected to peak at this wavelength, as observed for AdV [33,43], SARS-CoV-2 [44], and MS2 coliphages [15,43]. However, in our study, the greatest virucidal efficiency was observed at slightly longer wavelengths (U267–270). This shift was also reported in a recent UV-LED investigation [19]. Previous studies evaluating the UV action spectra for viral infectivity used various light sources, including tunable lasers, low-pressure mercury lamps, or combinations of lamps and UV-LEDs [15,33,43,44]. UV-LEDs typically have broader emission profiles (full width at half-maximum of approximately 10–15 nm) than the narrow monochromatic emission at U254 from low-pressure mercury lamps or tunable lasers [15,33,43,44], resulting in broader action spectra and a possible shift in the observed peak toward longer wavelengths. Therefore, to accurately establish UV action spectra for viral infectivity reduction, it is important to use the same type of light sources under standardized conditions, as performed in this study.

The structural characteristics of viral genomes might be important factors contributing to this shift. RNA secondary structures and differences in DNA/RNA photochemistry might also modulate the wavelength sensitivity of viruses [45]. In addition, because viral nucleic acids are packaged within nucleoprotein complexes or capsids, the surrounding proteins and biomolecules can influence the photochemical environment, potentially shifting the wavelength of maximum damage higher than U260 [33]. Although shorter wavelengths (e.g., U250–260) were effective for viral inactivation, their efficiency was slightly reduced, particularly for RSV and HMPV. These findings suggest that extremely short wavelengths might also be absorbed by viral proteins or other biomolecules, thereby reducing the targeting efficiency against nucleic acids alone.

In addition to wavelength-dependent effects, significant differences in UV-LED sensitivity were observed among the viral strains tested. Notably, RSV exhibited greater sensitivity to U280-LED irradiation than other enveloped RNA viruses such as IAV and CoV. Several factors might have contributed to this finding. First, RSV has a relatively small genome (approximately 15.2 kb) compared with SARS-CoV-2 (approximately 29.9 kb) or IAV (approximately 13.5 kb segmented across eight RNA segments), but importantly, the RSV genome consists of a single long, linear, negative-sense RNA strand [46]. Unlike segmented genomes (e.g., IAVs), a continuous genome might provide fewer opportunities for repair or functional redundancy following localized UV-induced damage, thus making RSV more vulnerable to genome disruption. Second, RSV virions possess a relatively simple nucleocapsid structure with less extensive protein shielding compared to CoVs, which have large, complex nucleoprotein layers surrounding their genomes [47]. This simpler structure could more directly expose the genomic RNA in RSV to UV irradiation, increasing the likelihood of photochemical damage. Third, RSV lacks robust innate repair mechanisms that could mitigate UV-induced genome damage after infection [48]. By contrast, certain large DNA viruses (e.g., HSV-1) or viruses with extensive replication machinery can tolerate or bypass UV lesions more effectively [49]. These combined structural and genomic characteristics likely render RSV especially vulnerable to UV-induced nucleic acid damage, accounting for its higher sensitivity to UV-LED disinfection observed compared to other enveloped viruses in this study.

In addition to the genomic structure, the presence or absence of a viral envelope influenced UV-LED sensitivity among the tested viruses. Our results demonstrated that nonenveloped viruses, specifically FCV and AdV-5, exhibited significantly higher resistance to UV-LED irradiation than enveloped RNA viruses. This finding is consistent with those of previous studies reporting that nonenveloped viruses tend to be more resilient to physical and chemical disinfection processes, including UV exposure [50]. The viral envelope, which is primarily composed of lipids, is relatively fragile and highly susceptible to oxidative stress and photochemical disruption. Consequently, envelope damage induced by UV irradiation can rapidly compromise the infectivity of enveloped viruses, even at moderate UV doses [51]. Contrarily, nonenveloped viruses rely on robust protein capsids to maintain environmental stability, and their inactivation primarily depends on nucleic acid damage, which often requires substantially higher UV fluences [52]. These observations illustrate that viral structural characteristics, particularly the presence or absence of an envelope, play critical roles in determining UV susceptibility. Furthermore, Western blotting did not detect significant viral protein degradation following UV-LED irradiation across all wavelengths tested, even under conditions in which substantial infectivity reduction was observed. This observation agrees with previous studies revealing that UV-induced protein degradation requires much higher fluence than genomic damage [25,53,54]. Together, our results indicate that UV-LED-mediated viral inactivation is predominantly driven by direct damage to viral genomes and that the practical optimal disinfection wavelength range lies around U267–270, not precisely at U260 as traditionally assumed. However, far-UVC radiation (U200–230) might more effectively induce protein damage in microorganisms [55]. Accurate action spectra across a wider wavelength range should be evaluated once high-power far-UVC-based LEDs become available in the future.

In this study, we demonstrated that the loss of infectivity following UV-LED irradiation was associated with damage to viral RNA, as confirmed by RT-qPCR, consistent with previous findings in poliovirus-1, SARS-CoV-2, and bacteriophages [21,24,25,56]. However, the specific types of RNA damage were not defined in those studies or in ours. It is well established that pyrimidines are more susceptible to UV-induced damage than purines [57]. UV irradiation generates various nucleic acid photoproducts. In DNA, these include CPDs, 6-4 pyrimidine–pyrimidone photoproducts (6-4PPs), and cytosine/thymine hydrates [57]. Similarly, RNA-specific photoproducts such as uracil cyclobutane dimers and uracil/cytosine hydrates have also been reported following UV exposure [58]. In our experiments, we confirmed the formation of CPDs in HSV-1 DNA following U250–U290-LED irradiation using dot-blot analysis with anti-CPD antibodies. Wurtmann and Wolin further reported that UV-induced photoproducts, including CPDs and 6-4PPs, can suppress both DNA and RNA synthesis [59]. These findings suggest that the reduction in RT-qPCR signal observed in our study likely results from the formation of such RNA photoproducts, which interfere with reverse transcription or PCR amplification.

We also assessed UV-LED-induced damage to viral proteins using Western blotting. However, Western blotting is a semi-quantitative method limited to detecting only epitope-preserving protein bands. It does not reflect total viral protein expression, and UV-induced structural changes, oxidative modifications, or protein aggregation may impair antibody recognition, even when protein expression levels are unchanged. Therefore, the reductions in band intensity observed after UV-LED irradiation may reflect structural damage to viral proteins rather than decreased synthesis. A previous study reported that the HSV-1 ICP0 protein was reduced following high-dose irradiation with U280-, U310-, and U365-LEDs [25]. Additionally, UVB and UVA irradiation have been shown to induce protein modifications in bacterial systems. For example, Bosshard et al. demonstrated that UVA irradiation caused oxidative damage and aggregation of proteins in *Escherichia coli* [60], and Santos et al. reported that UVB exposure led to oxidative modification of aromatic amino acid residues such as tyrosine and tryptophan in proteins from *Pseudomonas* sp. [61]. These findings support the possibility that UV-LED irradiation may damage viral proteins in ways that reduce their detectability by specific antibodies in immunoblotting assays, even when expression is not significantly altered.

Furthermore, in this study we conducted correlation analyses between infectivity reduction and viral genome or protein damage using data from five representative enveloped viruses (Table 3). While we acknowledge the value of broader pooled analyses based on viral categories (e.g., enveloped vs. nonenveloped and RNA vs. DNA), we limited our analysis to these five enveloped viruses to ensure experimental consistency in UV-LED exposure conditions and host cell systems, as well as to avoid variability that could confound interpretation. Previous studies have suggested that envelope status and the genome type can influence susceptibility to UV disinfection. For example, enveloped viruses tend to be more susceptible to environmental stressors, including UV light, likely due to the fragility of the lipid envelope [40]. In addition, a correlation between UV sensitivity and viral genome size or capsid structure has been observed in certain virus families [62]. However, Beck et al. demonstrated that for nonenveloped adenovirus type 2, the inactivation rate due to genome damage corresponded closely with the infectivity reduction at UV wavelengths of 253.7 nm, 270 nm, 280 nm, and 290 nm [43]. These findings underscore the complexity of UV-induced viral inactivation mechanisms and suggest that both genomic- and protein-level damage should be considered in future studies using broader virus panels and complementary detection techniques.

Finally, the findings of this study have important implications for practical disinfection strategies. Enveloped RNA viruses, including IAVs, CoVs, RSVs, and HMPV, which are major respiratory infectious agents, exhibited high sensitivity to UV-LED irradiation. These viruses spread efficiently via aerosol and fomite transmission routes, posing significant risks in healthcare, transportation, and public spaces [63]. The high UV susceptibility of these enveloped respiratory viruses suggests that UV-LED systems emitting near U267–270 could be effectively deployed for air disinfection, surface decontamination, and water treatment to mitigate viral transmission. Moreover, the wavelength-specific action spectra established in this study provide essential data for optimizing UV-LED system design, ensuring maximal inactivation efficacy while minimizing energy consumption [64].

By confirming that viral infectivity reduction is primarily related to nucleic acid damage rather than protein degradation, this work strengthens the mechanistic understanding required for developing targeted UV disinfection technologies, especially under the growing demand for environmentally friendly, mercury-free alternatives. Thus, our results contribute critical insights into the future deployment of UV-LED-based sterilization systems for controlling respiratory virus outbreaks and improving public health safety.

## 5. Conclusions

In this study, we determined the UV action spectra for viral infectivity reduction across representative respiratory and environmental viruses using a highly controlled UV-LED irradiation system. The results, as summarized in Figure 8, reinforce that viral nucleic acid damage is the dominant mechanism underlying UV-LED-mediated disinfection and suggest that UV-LED systems optimized for approximately U267–270 offer the highest virucidal efficacy. This integrated summary highlights the need to reconsider simplistic assumptions based solely on in vitro nucleic acid absorption when designing real-world UV disinfection systems. Future studies should further explore the contributions of viral genome features, such as structure and composition, to UV sensitivity and assess the scalability and practical deployment of UV-LED systems across the healthcare, food safety, and environmental sanitation sectors.

## Figures and Tables

**Figure 1 viruses-17-01065-f001:**
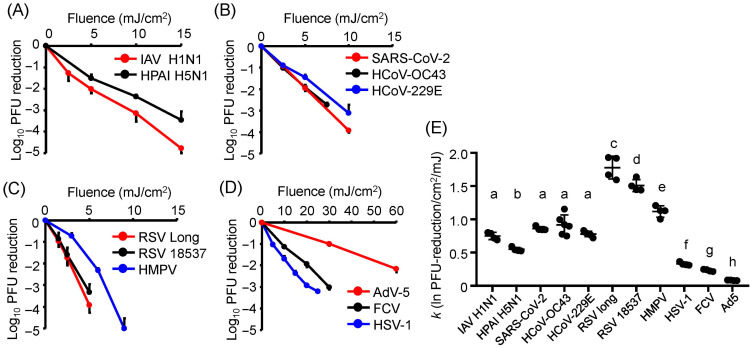
Reduction in viral infectivity by irradiation with U280 laser-emitting diodes (LEDs). (**A**–**D**) Infectivity reduction in each virus following irradiation with U280-LEDs. (**A**) Influenza A viruses (IAVs). (**B**) Coronaviruses (CoVs). (**C**) Respiratory syncytial viruses (RSVs) and human metapneumovirus (HMPV). (**D**) Herpes simplex virus type 1 (HSV-1), feline calicivirus (FCV), and adenovirus type 5 (AdV-5). (**E**) The inactivation rate constant (*k*) representing the infectivity reduction relative to the fluence of LED irradiation for each virus. Viral suspensions were irradiated with U280-LEDs at the indicated fluences and then used to infect host cells. Infectivity was measured by plaque-forming units (PFUs). The infectivity reduction is presented as the log_10_ PFU ratio. *k* was calculated as the ratio of the natural logarithm of the PFU reduction to the fluence of LED irradiation. Values are presented as the mean ± SD (*n* = 4–6). Different letters indicate statistically significant differences (*p* < 0.05) according to ANOVA followed by Bonferroni’s multiple comparison test.

**Figure 2 viruses-17-01065-f002:**
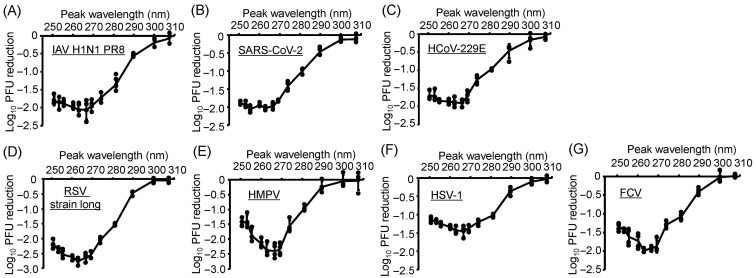
Ultraviolet action spectra of light-emitting diodes (LEDs) for infectivity reduction in viruses. (**A**) Influenza A virus (IAV) strain A/human/Puerto Rico/8/1934 (H1N1). (**B**) Severe acute respiratory syndrome coronavirus 2 (SARS-CoV-2) Hu/DP/Kng/19-020. (**C**) Human coronavirus (HCoV)-229E. (**D**) Respiratory syncytial virus (RSV) strain long. (**E**) Human metapneumovirus (HMPV) strain TN/83-1211. (**F**) Herpes simplex virus 1 (HSV-1) strain KOS. (**G**) Feline calicivirus (FCV) strain F-9. Viral suspensions were irradiated with LEDs at each peak wavelength at fluences of 2.5 (**A**,**B**), 3.0 (**C**), 2.0 (**D**), 4.0 (**E**), 5.0 (**F**), and 10.0 mJ/cm^2^ (**G**), corresponding to fluences that reduced infectivity by approximately 1 log_10_ under U280-LED irradiation. After irradiation, the suspensions were used to infect the host cells. Viral infectivity was measured by plaque-forming unit (PFU) assays. Infectivity reduction is presented as the log_10_ PFU ratio. Values are presented as the mean ± SD (*n* = 4–6).

**Figure 3 viruses-17-01065-f003:**
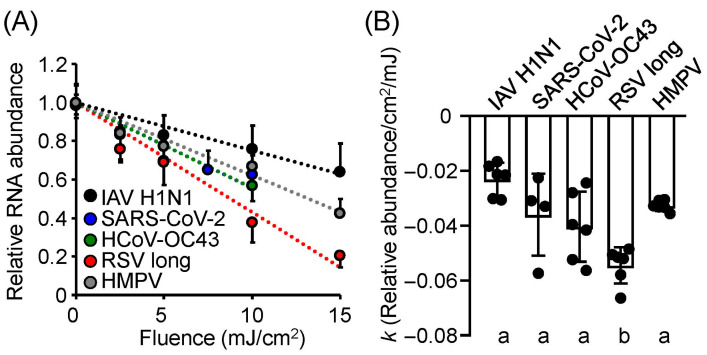
Damage to viral genomic RNA following irradiation with U280 light-emitting diodes (LEDs). (**A**) Relative RNA abundance of each virus following irradiation with U280-LEDs. (**B**) The inactivation rate constant (*k*) representing the reduction in viral RNA abundance relative to the fluence of LED irradiation for each virus. Viral suspensions were irradiated with U280-LEDs at the indicated fluences, and RNA was then extracted. Relative RNA abundance was measured by strand-specific RT-qPCR. *k* was calculated as the ratio of the reduction in the relative RNA abundance to the fluence of LED irradiation. Values are presented as the mean ± SD (*n* = 4–6). Different letters indicate statistically significant differences (*p* < 0.05) according to ANOVA followed by Bonferroni’s multiple comparison test.

**Figure 4 viruses-17-01065-f004:**
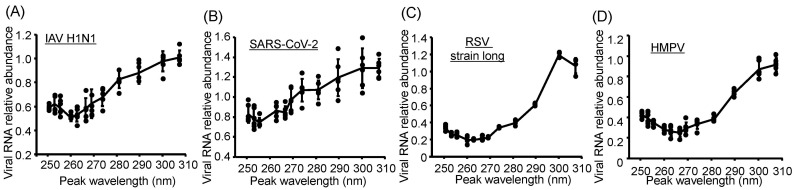
UV action spectra of light-emitting diodes (LEDs) for inducing viral RNA damage. (**A**) Influenza A virus (IAV) strain A/human/Puerto Rico/8/1934 (H1N1). (**B**) Severe acute respiratory syndrome coronavirus 2 (SARS-CoV-2) Hu/DP/Kng/19-020. (**C**) Respiratory syncytial virus (RSV) strain long. (**D**) Human metapneumovirus (HMPV) strain TN/83-1211. The viral suspensions were irradiated with LEDs at each peak wavelength at 10 (SARS-CoV-2 and RSV) or 15 mJ/cm^2^ (IAV and HMPV), and the viral RNA was then extracted. RNA damage was assessed using strand-specific RT-qPCR. Values are presented as the mean ± SD (*n* = 4–6).

**Figure 5 viruses-17-01065-f005:**
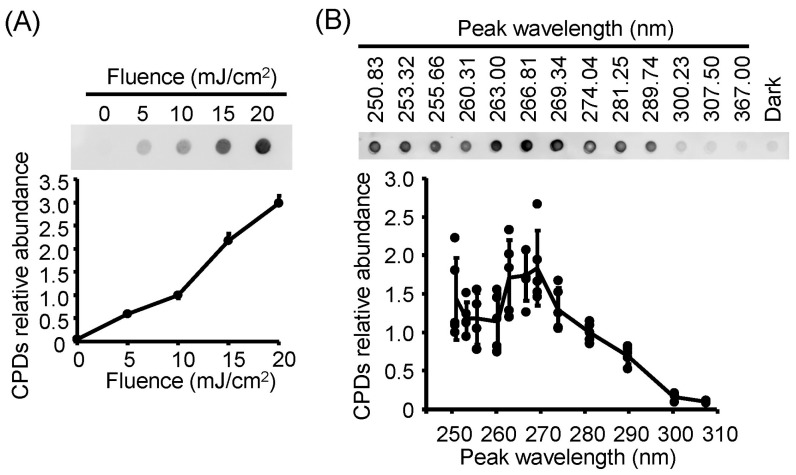
Viral DNA photoproducts of herpes simplex virus 1 (HSV-1 induced by irradiation with light-emitting diodes (LEDs). (**A**) Formation of cyclobutane pyrimidine dimers (CPDs), a DNA photoproduct, in HSV-1 DNA following irradiation with U280-LEDs at the indicated fluences. (**B**) UV action spectra of LEDs for CPD production in viral DNA. The top images present the representative results of immunoblotting for CPDs. The viral suspensions were irradiated with LEDs at each peak wavelength at a fluence of 10 mJ/cm^2^, and the viral DNA was then extracted. CPDs were detected by dot-blot analysis using an anti-CPDs antibody. Values are presented as the mean ± SD (*n* = 3–5).

**Figure 6 viruses-17-01065-f006:**
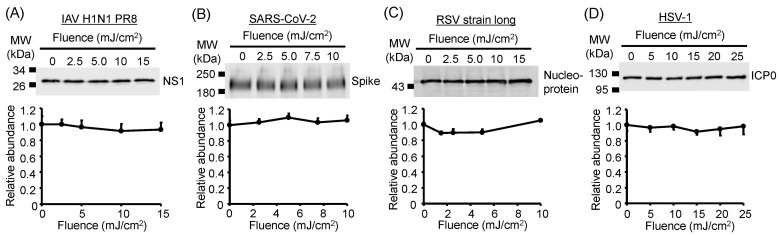
Damage to viral proteins by irradiation with U280 light-emitting diodes (LEDs). (**A**) Non-structural protein 1 (NS1) of influenza A virus (IAV) strain A/human/Puerto Rico/8/1934 (H1N1). (**B**) Spike glycoprotein of severe acute respiratory syndrome coronavirus 2 (SARS-CoV-2) Hu/DP/Kng/19-020. (**C**) Fusion glycoprotein of respiratory syncytial virus (RSV) strain long. (**D**) Infected cell protein 0 (ICP0) of herpes simplex virus-1 (HSV-1) strain KOS. The top images present the representative results of immunoblotting for each viral protein. Viral suspensions were irradiated with U280-LEDs at the indicated fluences, and proteins were then extracted. Viral proteins were detected by Western blotting using specific antibodies. Values are presented as the mean ± SD (*n* = 3–4).

**Figure 7 viruses-17-01065-f007:**
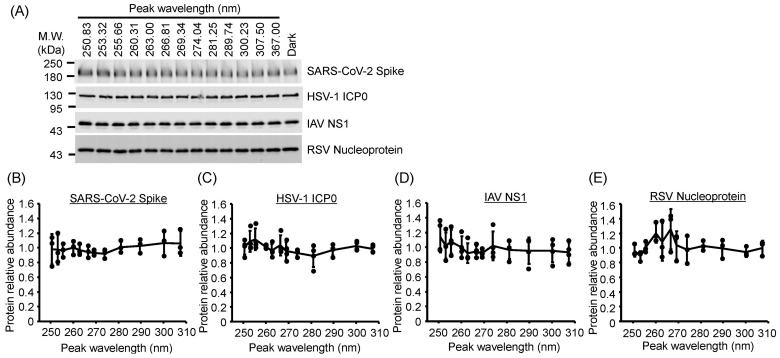
UV action spectra of light-emitting diodes (LEDs) for viral protein degradation. (**A**) Representative images of immunoblotting for viral proteins. (**B**–**E**) UV action spectra of LEDs for the degradation of (**B**) non-structural protein 1 (NS1) of influenza A virus (IAV) strain A/human/Puerto Rico/8/1934 (H1N1), (**C**) spike glycoprotein of severe acute respiratory syndrome coronavirus 2 (SARS-CoV-2) Hu/DP/Kng/19-020, (**D**) fusion glycoprotein of respiratory syncytial virus (RSV) strain long, and (**E**) infected cell protein 0 (ICP0) of herpes simplex virus-1 (HSV-1) strain KOS. The viral suspensions were irradiated with LEDs at each peak wavelength and a fluence of 10 (for SARS-CoV-2 and RSV) or 15 mJ/cm^2^ (for IAV and HSV-1), and the viral proteins were then extracted. Viral proteins were detected by Western blotting using specific antibodies. Values are presented as the mean ± SD (*n* = 3–4).

**Figure 8 viruses-17-01065-f008:**
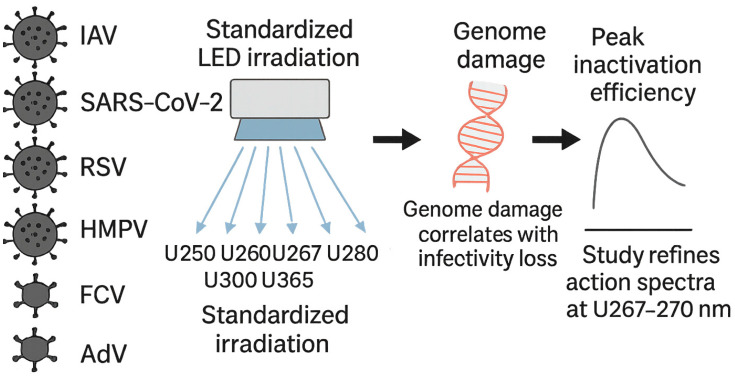
Graphical summary of wavelength-dependent viral inactivation by ultraviolet (UV) light-emitting diode (LED) irradiation (U250–365). Genome damage correlated with infectivity reduction across all tested viruses, with peak inactivation efficiency observed at U267–270. This study refines the wavelength-specific action spectra and highlights the dominant role of genomic damage in UV-LED-mediated viral disinfection. IAV, influenza A virus; SARS-CoV-2, severe acute respiratory syndrome coronavirus 2; RSV, respiratory syncytial virus; HMPV, human metapneumovirus; FCV, feline calicivirus; AdV-5, human adenovirus type 5.

**Table 1 viruses-17-01065-t001:** Virus and host cell strains.

Virus				Host Cells	
Envelope	Genome	Virus Strain	Provider	Cell Strain	Provider
Enveloped	ss-RNA	IAV strain A/Puerto Rico /8/1934 (H1N1)	Dr. Adachi ^1^	MDCK	ATCC
		HPAI strain A/crow /Kyoto/53/2004 (H5N1)	Dr. Nakaya ^2^	MDCK	ATCC
		SARS-CoV-2/Hu/DP /Kng/19-020	KPIPH	Vero E6 /TMPRSS2	JCRB
		HCoV-OC43	ATCC	Vero E6 /TMPRSS2	JCRB
		HCoV-229E	ATCC	HeLa /ACE2TMPRSS2	JCRB
		RSV strain long	ATCC	Hep-2	JCRB
		RSV strain CH18537	ATCC	Hep-2	JCRB
		HMPV strain TN/83-1211	BEI Resources	Hep-2	JCRB
	ds-DNA	HSV-1 strain KOS	ATCC	Vero	ATCC
Non- enveloped	ss-RNA	FCV strain F-9	ATCC	CRFK	ATCC
	ds-DNA	AdV-5 strain Adenoid 75	ATCC	Vero	ATCC

^1^ Tokushima University. ^2^ Kyoto Prefectural University of Medicine. ss-RNA, single-stranded RNA; ds-DNA, double-stranded DNA; IAV, influenza A virus; HPAI, highly pathogenic avian influenza virus; SARS-CoV-2, severe acute respiratory syndrome coronavirus 2; HCoV, human coronavirus; RSV, respiratory syncytial virus; HMPV, human metapneumovirus; HSV-1, herpes simplex virus 1; FCV, feline calicivirus; AdV-5, human adenovirus type 5; MDCK, Madin–Darby canine kidney cells; human embryonic kidney 293; CRFK, Crandell–Rees feline kidney cells; ACE2, angiotensin-converting enzyme 2; TMPRESS2, transmembrane protease serine 2; ATCC, American Type Culture Collection; KPIPH, Kanagawa Prefectural Institute of Public Health; JCRB, Japanese Collection of Research Bioresources.

**Table 2 viruses-17-01065-t002:** Comparison of UV fluence levels required to achieve 1–3 log_10_ reductions in viral infectivity between previous studies and the present study.

Virus Strain	Peak Wavelength of LEDs (nm)	Fluence (mJ/cm^2^)			Reference
		Log_10_ Infectivity Reduction			
		−1	−2	−3	
Human coronavirus OC43	279	3.5	5.5	7.0	Gerchman [40]
	281.3	2.7	5.2	7.7	This study
Human coronavirus 229E	282	3.7	7.4	11.1	Ma [16]
	281.3	3	5.0	9.7	This study
Severe acute respiratory syndrome coronavirus 2	282	1.9	3.8	5.7	Ma [19]
	281.3	2.0	5.0	7.5	This study
Feline calicivirus	281	9.0	18.9	28.9	Oguma [35]
	281.3	8.8	20.6	30.0	This study

**Table 3 viruses-17-01065-t003:** Pearson’s correlation coefficients (*r*) of the action spectra of ultraviolet light-emitting diodes.

Virus Strain	Action Spectra for Viral Infectivity (Figure 2)			
	Action Spectra for Viral Genomes (Figure 4)		Action Spectra for Viral Proteins (Figure 6)	
	*r*	*p*	*r*	*p*
Influenza A virus H1N1	0.966	<0.001	−0.319	0.312
Severe acute respiratory syndrome coronavirus 2	0.959	<0.001	0.750	0.008
Respiratory syncytial virus strain long	0.947	<0.001	−0.415	0.180
Human metapneumovirus	0.930	<0.001	−	−
Herpes simplex virus type 1	–0.937	<0.001	−0.0736	0.820

## Data Availability

The original contributions presented in this study are included in the article/Appendix A. Further inquiries can be directed to the corresponding author (K.M.).

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
