# Peer review of "Viral Inactivation by Light-Emitting Diodes: Action Spectra Reveal Genomic Damage as the Primary Mechanism"

_viruses, 2025, doi:10.3390/v17081065_

Round 1

Reviewer 1 Report

Comments and Suggestions for Authors

Comments

Line 13 – “reduction” used twice in same sentence, redundant, suggest deleting second occurrence. Also, is this ‘reduction’ induced by UV-LED exposure? If so (I assume so), all wavelengths? Need to tie this reduction to the cause (UV-LED exposure)

Line 140 – can you provide a reference for this ‘previous’ study?

Table 1 – it would be useful to provide the characteristics of each virus as a column, For example providing ‘enveloped’ or ‘non-enveloped’ and ‘ds-DNA’ or ss-RNA, or ss-DNA, characteristics of each.

Line 195 – how many replicates of each 10-fold dilution were plated/infected/analyzed?

Line 206 – was RT-qPCR used for all viruses? Even DNA viruses like adeno and HSV-1? Perhaps a sentence to indicate which viruses were subjected to this method would reduce confusion.

Line 216 – suggest indicating the purpose of the Dot-blot analysis, similar to how RT-qPCR section was started.. “Damage to viral DNA caused by UV irradiation was assessed using…”

Line 233 – western blot was performed only on one protein from each virus. There is a limitation in making broad conclusions about total protein expression. Also, could UV exposure impact antibody binding of proteins? If so, expression may be unimpacted but yet the protein that was produced could have been damaged by UV and show differential presence in Western blotting.

Figure 1 – there is no key for the ‘different letters indicate statistically significant differences’ levels, please provide

Table 3 – were these correlation analyses conducted on all viruses pooled, rather than pooled by like virus types? How about all enveloped pooled and all non-enveloped pooled? How about all DNA viruses pooled and all RNA viruses pooled? It could be interesting to see how these virus categories (more broadly) correlate with genome and protein data.

Line 498 – “prevent” protein damage or “induce” protein damage?? The paper cited documents antimicrobial impacts of 222 and lack of human skin impacts.

Figure 8 – I think ‘refines’ is misspelled

Author Response

Comments 1: Line 13 – “reduction” used twice in same sentence, redundant, suggest deleting second occurrence. Also, is this ‘reduction’ induced by UV-LED exposure? If so (I assume so), all wavelengths? Need to tie this reduction to the cause (UV-LED exposure).

Response 1: Thank you for your comment. We agree with your suggestion and have deleted the second occurrence of “reduction” at Line 30, Page 1 of the revised manuscript. Additionally, we clarified the cause by inserting “induced by UV-LED exposure” as you recommended.

Comments 2: Line 140 – can you provide a reference for this ‘previous’ study?

Response 2: Thank you for your comment. We have added appropriate references to the previous study at Line 143, Page 3 of the revised manuscript.

Comments 3: Table 1 – it would be useful to provide the characteristics of each virus as a column, For example providing ‘enveloped’ or ‘non-enveloped’ and ‘ds-DNA’ or ss-RNA, or ss-DNA, characteristics of each.

Response 3: Thank you for your suggestion. We agree and have added the characteristics of each virus—including envelope and genome type—to Table 1, Page 4 of the revised manuscript.

Comments 4: Line 195 – how many replicates of each 10-fold dilution were plated/infected/analyzed?

Response 4: Thank you for your comment. Each 10-fold serial dilution of irradiated or non-irradiated viral suspension was plated/infected/analyzed in duplicate in every experiment. We have added this information in Line 205, Page 5 of the revised manuscript.

Comments 5: Line 206 – was RT-qPCR used for all viruses? Even DNA viruses like adeno and HSV-1? Perhaps a sentence to indicate which viruses were subjected to this method would reduce confusion.

Response 5: Thank you for your comment, and we apologize for the confusion. RT-qPCR was performed only for the four RNA viruses: influenza A virus (H1N1 subtype), SARS-CoV-2, and RSV (strain long), and HMPV. We have added this clarification in Line 217, Page 5 of the revised manuscript.

Comments 6: Line 216 – suggest indicating the purpose of the Dot-blot analysis, similar to how RT-qPCR section was started.. “Damage to viral DNA caused by UV irradiation was assessed using…”

Response 6: Thank you for your suggestion. We have added the sentence “Damage to HSV-1 viral DNA caused by UV irradiation was assessed using dot-blot analysis” at Line 228, Page 6 of the revised manuscript.

Comments 7: Line 233 – western blot was performed only on one protein from each virus. There is a limitation in making broad conclusions about total protein expression. Also, could UV exposure impact antibody binding of proteins? If so, expression may be unimpacted but yet the protein that was produced could have been damaged by UV and show differential presence in Western blotting.

Response 7: Thank you for your insightful comment. We acknowledge that only one representative viral protein was analyzed per virus in the Western blot experiments, which limits our ability to draw broad conclusions about total viral protein expression. We have added a sentence to the Discussion section (Line 535, Page 14 of the revised manuscript) to acknowledge this as a limitation of the study.

Additionally, you raise an important point regarding the potential impact of UV exposure on antibody binding affinity. Western blotting is a semi-quantitative method limited to detecting only epitope-preserving protein bands. It does not reflect total viral protein expression, and UV-induced structural changes, oxidative modifications, or protein aggregation may impair antibody recognition, even when protein expression levels are unchanged. Therefore, the reductions in band intensity observed after UV-LED irradiation may reflect structural damage to viral proteins rather than decreased synthesis. A previous study reported that the HSV-1 ICP0 protein was reduced following high-dose irradiation with U280-, U310-, and U365-LEDs [25]. Additionally, UVB and UVA irradiation have been shown to induce protein modifications in bacterial systems. For example, Bosshard et al. demonstrated that UVA irradiation caused oxidative damage and aggregation of proteins in Escherichia coli [63], and Santos et al. reported that UVB exposure led to oxidative modification of aromatic amino acid residues such as tyrosine and tryptophan in proteins from Pseudomonas sp. [64]. These findings support the possibility that UV-LED irradiation may damage viral proteins in ways that reduce their detectability by specific antibodies in immunoblotting assays, even when expression is not significantly altered. We have added this point to the Discussion section, Page 14 of the revised manuscript.

Comments 8: Figure 1 – there is no key for the ‘different letters indicate statistically significant differences’ levels, please provide.

Response 8: Thank you for your comment. The figure legend already included the explanation “Different letters indicate statistically significant differences (P < 0.05) according to ANOVA followed by Bonferroni’s multiple comparison test.” Additionally, the letters indicating significance are shown directly on the relevant data plots in Figure 1E. We believe this now provides sufficient clarity for interpretation.

Comments 9: Table 3 – were these correlation analyses conducted on all viruses pooled, rather than pooled by like virus types? How about all enveloped pooled and all non-enveloped pooled? How about all DNA viruses pooled and all RNA viruses pooled? It could be interesting to see how these virus categories (more broadly) correlate with genome and protein data.

Response 9: Thank you for your thoughtful comment. In the present study, we conducted correlation analyses using data from five representative enveloped viruses: influenza A virus (IAV), SARS-CoV-2, RSV, HMPV, and HSV-1. While we acknowledge the value of broader pooled analyses based on viral categories (e.g., enveloped vs. non-enveloped, RNA vs. DNA), we limited our analysis to these five enveloped viruses to ensure experimental consistency in UV-LED exposure conditions and host cell systems, and to avoid variability that could confound interpretation. Previous studies have suggested that envelope status and genome type can influence susceptibility to UV disinfection. For example, enveloped viruses tend to be more susceptible to environmental stressors, including UV light, likely due to the fragility of the lipid envelope [65]. In addition, a correlation between UV sensitivity and viral genome size or capsid structure has been observed in certain virus families [66]. However, Beck et al. demonstrated that for non-enveloped adenovirus type 2, the inactivation rate due to genome damage corresponded closely with the infectivity reduction at UV wavelengths of 253.7 nm, 270 nm, 280 nm, and 290 nm [41]. These findings underscore the complexity of UV-induced viral inactivation mechanisms and suggest that both genomic and protein-level damage should be considered in future studies using broader virus panels and complementary detection techniques. We have noted this as a potential direction in the Discussion section, Page 14-15 of the revised manuscript.

Comments 10: Line 498 – “prevent” protein damage or “induce” protein damage?? The paper cited documents antimicrobial impacts of 222 and lack of human skin impacts.

Response 10: Thank you for your comment, and we apologize for the confusion caused by the incorrect use of the word “prevent.” We have revised the sentence to: “far-UVC radiation (200–230 nm) may more effectively induce protein damage in microorganisms.” This correction appears in Line 515, Page 14 of the revised manuscript.

Comments 11: Figure 8 – I think ‘refines’ is misspelled

Response 11: Thank you for your comment, and we apologize for the misspelling. We have corrected the error in the revised manuscript.

Reviewer 2 Report

Comments and Suggestions for Authors

The submitted manuscript (viruses-3699813) describes the results of a mechanistic study concerning the effect of UV wavelength on the efficiency of virus degradation. This study is well designed and provides interesting results that are important for practical application of UV-assisted decontamination methods. The several concerns should be addressed before acceptance of this manuscript for publication in the Virusesjournal:

  1. I think that photocatalysis should be also mentioned in the Introduction among other approaches for virus degradation (LL. 72-75), because there are many published papers showing complete degradation of viruses on the UV- or Vis-irradiated surface of photocatalytic materials and coatings (e.g., self-cleaning photocatalytic textiles).
  2. In additions to fluence rates (Table S1), time periods of treatments with each UV-LED are to be mentioned in the experimental section for clearer conversion of mW/cm2 to mJ/cm2 units.
  3. The mechanism of NA degradation under exposure to UV should be deeply discussed. Namely, what are the types of damages (single-strand or/and double-strand breaks, degradation of nitrogenous) that provide genomic NA degradation and reduction in their signal in the PCR method?
  4. Typos error in L.270.

Author Response

Comments 1: I think that photocatalysis should be also mentioned in the Introduction among other approaches for virus degradation (LL. 72-75), because there are many published papers showing complete degradation of viruses on the UV- or Vis-irradiated surface of photocatalytic materials and coatings (e.g., self-cleaning photocatalytic textiles).

Response 1: Thank you for your suggestion. We have added the sentence “Photocatalytic effects induced by UV- or visible-light irradiation of photocatalytic materials (e.g., TiOâ‚‚) primarily degrade viral proteins” at Line 82, Page 2 of the revised manuscript.

Comments 2: In additions to fluence rates (Table S1), time periods of treatments with each UV-LED are to be mentioned in the experimental section for clearer conversion of mW/cm2 to mJ/cm2 units.

Response 2: Thank you for your suggestion. We have added a sentence describing the time periods of treatment with each UV-LED at the end of Section 2.2, Page 5 of the revised manuscript to clarify the conversion from mW/cm² to mJ/cm²

Comments 3: The mechanism of NA degradation under exposure to UV should be deeply discussed. Namely, what are the types of damages (single-strand or/and double-strand breaks, degradation of nitrogenous) that provide genomic NA degradation and reduction in their signal in the PCR method?

Response 3: Thank you for your suggestion. In this study, we demonstrated that the loss of infectivity following UV-LED irradiation was associated with damage to viral RNA, as confirmed by RT-qPCR, consistent with previous findings in poliovirus-1, SARS-CoV-2, and bacteriophages [20, 24, 25, 59]. However, the specific types of RNA damage were not defined in those studies or in ours. It is well established that pyrimidines are more susceptible to UV-induced damage than purines [60]. UV irradiation generates various nucleic acid photoproducts. In DNA, these include CPDs, 6-4 pyrimidine–pyrimidone photoproducts (6-4PPs), and cytosine/thymine hydrates [60]. Similarly, RNA-specific photoproducts such as uracil cyclobutane dimers and uracil/cytosine hydrates have also been reported following UV exposure [61]. In our experiments, we confirmed the formation of CPDs in HSV-1 DNA following U250–U290-LED irradiation using dot-blot analysis with anti-CPD antibodies. Wurtmann and Wolin further reported that UV-induced photoproducts, including CPDs and 6-4PPs, can suppress both DNA and RNA synthesis [62]. These findings suggest that the reduction in RT-qPCR signal observed in our study likely results from the formation of such RNA photoproducts, which interfere with reverse transcription or PCR amplification.

We have incorporated this discussion of the mechanism of viral nucleic acid degradation under UV exposure into the Discussion section, Page 14 of the revised manuscript.

Comments 4: Typos error in L.270.

Response 4: Thank you for your comment, and we apologize for any typographical errors. We carefully reviewed the text around Line 270 in the original manuscript but could not identify any specific typographical mistakes at that location. However, we acknowledge that other typographical errors were pointed out by another reviewer. We have thoroughly checked the entire manuscript and corrected those errors in the revised version.